# An Image Is Worth Ten Thousand Words: Verbose-Text Induction Attacks on VLMs

## Abstract

With the remarkable success of Vision-Language Models (VLMs) on multimodal tasks, concerns regarding their deployment efficiency have become increasingly prominent. In particular, the number of tokens consumed during the generation process has emerged as a key evaluation metric. Prior studies have shown that specific inputs can induce VLMs to generate lengthy outputs with low information density, which significantly increases energy consumption, latency, and token costs. However, existing methods simply delay the occurrence of the EOS token to *implicitly* prolong output, and fail to directly maximize the output token length as an *explicit* optimization objective, lacking stability and controllability. To address these limitations, this paper proposes a novel verbose-text induction attack (VTIA) to inject imperceptible adversarial perturbations into benign images via a two-stage framework, which identifies the most malicious prompt embeddings for optimizing and maximizing the output token of the perturbed images. Specifically, we first perform *adversarial prompt search*, employing reinforcement learning strategies to automatically identify adversarial prompts capable of inducing the LLM component within VLMs to produce verbose outputs. We then conduct *vision-aligned perturbation optimization* to craft adversarial examples on input images, maximizing the similarity between the perturbed image's visual embeddings and those of the adversarial prompt, thereby constructing malicious images that trigger verbose text generation. Comprehensive experiments on four popular VLMs demonstrate that our method achieves significant advantages in terms of effectiveness, efficiency, and generalization capability.

## 1 Introduction

In recent years, Vision-Language Models (VLMs) have advanced rapidly, demonstrating strong performance in multimodal tasks such as image captioning (Wang et al., 2021; Li et al., 2022), visual question answering (Liu et al., 2023b; Dai et al., 2023), and visual reasoning (Zhu et al., 2023; Cheng et al., 2024). However, these achievements are largely driven by scaling models to billions of parameters, which renders the model's deployment highly resource-intensive (Zhang et al., 2024a). With the accelerating commercialization of VLMs, many service providers offer inference services for users through APIs, commonly under token-based billing schemes. This raises a critical challenge for practical deployment, which is to ensure accurate and efficient inference while suppressing redundant token generation.

Most state-of-the-art VLMs adopt modular architectures in which an intermediate layer connects the visual encoder and Large Language Models (LLMs) (Yin et al., 2024). The visual encoder and intermediate layer jointly process the image to extract visual embeddings, which are then combined with the textual embeddings of prompts for response generation by the LLMs. Due to the autoregressive nature of LLMs, each generated token requires a forward pass, and the token is subsequently fed back as input for the next steps (Cao et al., 2023). As the token sequence grows, the computational cost of each forward pass increases accordingly. Such characteristics expose VLMs to a new type of security threat (Gao et al., 2024a;b): adversaries can design malicious images that induce the model to generate excessively long outputs. Such behavior not only inflates inference energy consumption and latency, severely degrading server responsiveness, but also substantially increases user costs under token-based pricing. As illustrated in Figure 1, the inference time increases with the number

of generated tokens. When adversarial perturbations remain visually imperceptible, the stealth and practical risk of these attacks become even more severe.

Prior works (Shumailov et al., 2021; Chen et al., 2022b) have investigated increasing inference energy consumption and latency by adding perturbations to images, but these methods mainly target image-classification models (e.g., ResNet) or small-scale image-to-text models (e.g., ResNet+RNN) and do not readily transfer to modern VLMs. Recent studies (Gao et al., 2024a;b) on VLMs have focused on prolonging outputs by delaying the occurrence of the EOS token: their core idea is to decrease the probability of EOS in the next-step distribution and use that signal to compute gradients for optimizing image perturbations. However, this approach relies solely on the probability distribution obtained from a single forward pass of the image and input text through the VLM, therefore cannot capture the complete information

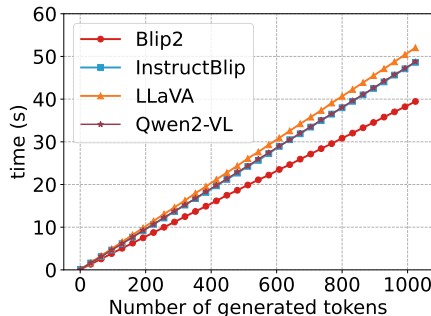

Figure 1: The relationship between the time consumed in a single inference and the number of generated tokens.

of the subsequent autoregressive generation process. That is because LLMs generate autoregressively, later outputs are highly context-dependent and thus difficult to predict or control. Consequently, adversarial images optimized using single-pass information often lack stability and controllability in their final attack effectiveness. This limitation raises a key question: **can we directly use the VLM's output length as the optimization objective when optimizing an adversarial image for verbose text, thereby improving the stability and controllability of adversarial methods?**

To address these limitations, in this paper, we propose a novel redundancy-inducing VLM attack, termed Verbose-Text Induction Attack (VTIA). This attack method adopts a two-stage decoupling strategy that explicitly learn the most malicious prompt embedding and maximizes the output token numbers of the perturbed images. In particular, it proceeds in two steps: 1) Adversarial Prompt Search: we train an attacker LLM using reinforcement learning to optimize the generation of a malicious prompt, avoiding the non-differentiability of directly maximizing output token length. The embedding of this prompt, when inserted after the visual embeddings, can trigger the LLM within the VLM to produce excessively long outputs; 2) Vision-Aligned Perturbation Optimization: based on the similarity between the malicious prompt embedding and visual embeddings, gradients are computed to perturb the input image and obtain adversarial examples. This stage operates entirely independently of the target VLM's textual module, thereby avoiding the substantial overhead of repeatedly invoking large LLMs during iterative optimization. In this manner, our attack can effectively prolong the VLM's output. The main contributions of this work are as follows:

- We propose a novel verbose-text induction attack on VLMs, capable of generating adversarial images while accounting for subsequent outputs with explicit token-aware designs, thereby advancing security research on inducing verbose text generation in VLMs.
- We design a two-stage attack framework, which firstly searches for an adversarial prompt through reinforcement learning, and then uses it to optimize adversarial images with the defined similarity loss and standard deviation loss.
- We apply our method to four mainstream VLMs (Blip2, InstructBlip, LLaVA, Qwen2-VL) and evaluate it on the MS-COCO dataset. Experimental results show that the generated adversarial images can induce these models to produce token counts that are 121.90×, 87.19×, 9.44×, and 6.48× longer than those generated from the original images.

## 2 RELATED WORK

### 2.1 VLMS

Currently, mainstream VLMs consist of two key parts, i.e., textual and visual components. Early models such as CLIP (Radford et al., 2021), BLIP (Li et al., 2022), and ALIGN (Jia et al., 2021) employed both visual encoders and text encoders, aligning image and text embeddings through

contrastive learning. Newer generations of models (e.g., Blip2 (Li et al., 2023), InstructBlip (Dai et al., 2023), MiniGPT (Zhu et al., 2023), LLaVA (Liu et al., 2023b), Qwen2-VL (Wang et al., 2024)) typically no longer include a standalone text encoder. Instead, they rely on LLMs, such as OPT (Zhang et al., 2022), LLaMA (Touvron et al., 2023), Vicuna (Chiang et al., 2023), and Qwen (Bai et al., 2023), for text understanding, while integrating visual inputs through projection layers or cross-attention mechanisms. This trend reflects a growing shift toward leveraging the capabilities of LLMs, rather than relying solely on visual components, to support more flexible and advanced multimodal reasoning and generation tasks.

## 2.2 ENERGY-LATENCY ATTACKS

Prior research (Chen et al., 2022a; Hong et al., 2020; Liu et al., 2023a; Chen et al., 2023; Zhang et al., 2024b; Dong et al., 2024) has investigated how to construct adversarial inputs to degrade the model inference efficiency. Shumailov et al. (2021) analyzed both language and vision models; in the case of vision models, the focus was on classification architectures such as ResNet (He et al., 2016), DenseNet (Huang et al., 2017), and MobileNet (Howard et al., 2017). The approach involved designing adversarial image inputs that increase activation values across layers. Higher activation density prevents hardware from skipping certain computations, thereby increasing energy consumption. However, this work did not consider multimodal models. Chen et al. (2022b) examined the efficiency of Neural Image Caption Generation (NICG) models, proposing to delay the occurrence of EOS tokens while disrupting token dependencies, thereby generating longer sequences. This increases the number of decoder calls and reduces inference efficiency. Nonetheless, their studied architectures (MobileNets+LSTM, ResNet+RNN) differ significantly from the Transformer-based architectures used in current mainstream VLMs. To induce VLMs to generate longer responses, Gao et al. (2024a) and Gao et al. (2024b) proposed three strategies: 1) lowering the probability of EOS token generation to delay its appearance; 2) enhancing output uncertainty to encourage predictions that deviate from the original token order and pay more attention to alternative candidate tokens; and 3) improving the diversity of hidden states across generated tokens to explore a broader output space, thereby further weakening original output dependencies. However, these works typically proxy increased output verbosity by manipulating the EOS token probability rather than treating token length as an explicit optimization objective. Given the autoregressive nature of current models, where outputs serve as inputs for subsequent steps, and the fact that the loss function is constructed solely from distributions obtained in a single forward pass, the attack effectiveness of such adversarial samples remains difficult to guarantee.

## 3 PRELIMINARIES

### 3.1 STRUCTURE OF VLMS

Existing state-of-the-art VLMs, such as Blip2 (Li et al., 2023), InstructBLIP (Dai et al., 2023), LLaVA (Liu et al., 2023b), and Qwen2-VL (Wang et al., 2024), generally consist of a visual encoder $\mathcal{E}$ and a pretrained LLM $\mathcal{F}$. To bridge the two components, an intermediate module $\mathcal{M}$ is required. For example, in InstructBLIP, this module consists of a Q-Former and a fully connected layer. While in LLaVA, it is implemented as a linear layer that maps the visual features extracted by the visual encoder into the word embedding space.

Given an input image $x$, the visual encoder first encodes the input image as visual features $Z_v = \mathcal{E}(x)$. Subsequently, the intermediate module projects the visual features into visual embeddings $H_v = \mathcal{M}(Z_v)$, which has the dimension of $m$ (i.e., the visual token number of the VLM). And for the input prompt $c$, it is first processed by tokenizer $\mathcal{T}$ into a textual token sequence $S_t = \mathcal{T}(c) = \{s_1, s_2, \cdots, s_n\}$ of length $n$. Then $S_t$ is projected by the embedding layers $\mathcal{D}$ into textual embeddings $H_t = \mathcal{D}(S_t)$. Finally, the visual embedding $H_v$ is concatenated with the textual embedding $H_t$ to form the initial sequence, and then fed into the LLM for content generation in an autoregressive manner. Represent the initial sequence as $H_v \oplus H_t$, it is fed into the LLM $\mathcal{F}$, which produces a probability distribution over the next token. By sampling from this distribution, the next token is obtained and appended to the original sequence, which serves as the input for the next decoding step of the LLM. Formally, the response of the LLM can be denoted as $y = \mathcal{F}(\mathcal{M}(\mathcal{E}(x) \oplus \mathcal{D}(\mathcal{D}(c)))$. The generation process terminates under either of the following

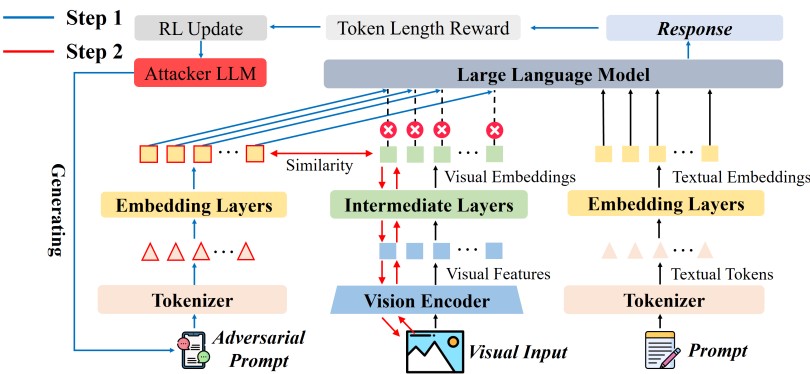

Figure 2: Flowchart of VTIA. Step 1: Adversarial prompt search; Step 2: Vision-aligned perturbation optimization.

conditions: 1) The generated token is an EOS token in a given step. 2) The number of generated tokens reaches a predefined maximum value.

## 3.2 THREAT MODEL

**Attacker's Knowledge.** We consider a gray-box attack setting in which the attacker has access to the model structure of the target VLM $f$, as well as the parameters of the visual encoder $\mathcal{E}$ and the intermediate module $\mathcal{M}$. While the attacker does not require the LLM's parameters.

**Attacker's Goal.** The attacker aims to generate an adversarial image that induces the VLM to produce maximally verbose responses. Such responses increase inference costs, including computational and energy consumption, latency, and monetary expenses.

**Attacker's Constraint.** The magnitude of the perturbations applied to the image is bounded by a predefined $l_p$ norm, ensuring the stealthiness of the attack.

## 4 ATTACK METHOD

### 4.1 INSIGHT OF VTIA

The goal of our attack is to find an adversarial perturbation $\delta$ that, when added to a clean image $x$, yields a perturbed image $x^* = x + \delta$ that causes the victim VLM $f$ to produce the output $y$ with maximal token length. Formally, let $\texttt{len}(\cdot)$ denotes the token-count operator and let $f$ represents the target VLM, we aim to solve

$$\max_{\delta} \mathbb{E}_{y=\mathcal{F}(\mathcal{M}(\mathcal{E}(x^*)\oplus\mathcal{D}(\mathcal{D}(c))} \left[\texttt{len}(y)\right], \tag{1}$$

$$\text{s.t. } ||x^* - x||_p \leq \epsilon, \tag{2}$$

where $||\cdot||_p$ is the $l_p$ norm constraint and $\epsilon$ indicates the perturbation magnitude. However, the above problem cannot be solved directly because $\texttt{len}(y)$ is not differentiable with respect to $\delta$. Therefore, we design two steps to achieve the attack goal: 1) **Adversarial prompt search**: We directly construct the token length of the VLM's response as the reinforcement learning reward. To reduce the search space, we optimize an attacker LLM to produce discrete textual prompts whose embeddings replace image embeddings, thereby inducing the targeted adversarial behavior. 2) **Vision-aligned perturbation optimization**: We split the optimized adversarial prompt into token slices and optimize an objective that jointly penalizes slice–image embedding dissimilarity and standard deviation, and apply backpropagation to optimize and obtain the adversarial image. Our proposed VTIA can capture the VLM's output during the adversarial prompt search stage, compensating for the limitation of existing approaches (Gao et al., 2024a;b), which cannot observe the subsequent autoregressive generation process when creating adversarial images. Figure 2 illustrates the workflow of our proposed attack method.

---

**Algorithm 1** Process of vision-aligned perturbation optimization

---

1: **Input:** Origin images $x$, the perturbation magnitude $\epsilon$, step size $lr$, optimization iterations $T$ and momentum value $\mu$;
2: **Output:** An adversarial image $x^*$ with $\|x^* - x\|_p \leq \epsilon$.
3: $g_0 = 0$, $x_0^* = x$;
4: **for** $t = 0$ to $T - 1$ **do**
5:     Input $x_t^*$ to VLM and calculate the loss $\mathcal{L}_{total}$ according to Equation (8);
6:     Update $g_{t+1}$ by:

$$g_{t+1} = \mu \cdot g_t + \frac{\nabla \mathcal{F}(x_t^*)}{\|\nabla \mathcal{F}(x_t^*)\|_1}; \tag{4}$$

7:     Update $x_{t+1}^*$ by:

$$x_{t+1}^* = x_t^* - lr \cdot sign(g_{t+1}); \tag{5}$$

8: **end for**
9: **return** $x_T^*$

---

## 4.2 ADVERSARIAL PROMPT SEARCH

In the first step, we optimize an attacker LLM $\mathcal{F}^*$ to produce adversarial prompts $c^*$. Then $c^*$ is tokenized and projected into textual embeddings $H_t^*$, and is used to replace the visual embedding $H_v$ of the target VLM. The search objective is to maximize the VLM's output length (i.e., induce the most verbose responses). This problem can be naturally solved through the following formulation of reinforcement learning:

$$\arg\max_{\mathcal{F}^*} \mathbb{E}_{y = \mathcal{F}(\mathcal{D}(\mathcal{T}(c^*)) \oplus \mathcal{D}(\mathcal{T}(c)))}[\texttt{len}(y)], \tag{3}$$

which takes the token length of the response as the reward. We use $\mathcal{F}^*$ to generate an adversarial prompt $c^*$ containing $k$ tokens, and slice its corresponding textual embedding $H_t^*$ according to the visual token number $m$ corresponding to the target VLM. Specifically, we set the dimension corresponding to the sliced embedding $H_t^*[k']$ to an integer $k'$ that is divisible by $m$ (e.g., when $m$ is 32, $k'$ can be 4), corresponding to the vector of the first $k'$ dimensions of $H_t^*$. Subsequently, we repeat $H_t^*[: k']$ for $m/k'$ times and replace it with the model's visual embeddings to generate the response. We use the Proximal Policy Optimization (PPO) strategy to optimize $f^*$ according to the objectives in Equation (3). By repeating this process, one can eventually identify an adversarial prompt that induces the LLM to generate a token count reaching the predefined upper bound.

## 4.3 VISION-ALIGNED PERTURBATION OPTIMIZATION

In order to get the adversarial image $x^*$, we optimize the perturbation $\delta$ through vision-aligned perturbation optimization based on the generated adversarial prompt $c^*$. Let the visual embedding of the adversarial image be represented as $H_v^* = [v_1, v_2, v_3, \ldots, v_m]$, where $m$ denotes the number of visual tokens, and $v_i$ represents the visual embedding vector. Correspondingly, the concatenated embedding of the adversarial prompt slice is represented as $[H_t^*[: k']_1, H_t^*[: k']_2, \cdots, H_t^*[: k']_{m/k'}] = [t_1, t_2, t_3, \ldots, t_m]$, where $t_i$ denotes the text embedding vector. Since the concatenated adversarial prompt embedding is fixed after step one, we need to optimize a perturbation $\delta$, so that the adversarial image's embedding closely matches the prompt's per-token embeddings, thereby reproducing the same verbose behavior. Therefore, it is necessary to maximize the cosine similarity between $v_i$ and $t_i$. Upon this, we define the similarity loss $\mathcal{L}_{sim}$ as:

$$\mathcal{L}_{sim} = \frac{1}{m} \sum_{i=1}^{m} \cos(\mathcal{M}(\mathcal{E}(x + \delta)[i], t_i), \tag{6}$$

which is the mean cosine similarity between the visual and textual embeddings. However, if only $\mathcal{L}_{sim}$ is used as the loss term, the optimization process may lead to a situation where some $(v_i, t_i)$ pairs achieve sufficient optimization, while others $(v_j, t_j)$ remain under-optimized. This imbalance can result in adversarial images with suboptimal attack performance. To address this issue and ensure that each $(v_i, t_i)$ pair is adequately optimized, we introduce a standard deviation term into

Table 1: Key information of the large models used in the experiments, including model scale (number of parameters), type of visual module, the LLM employed, and the number of visual tokens.

| Model | Parameters | Vision Encoder | LLM | Visual token number |
|-------|-----------|----------------|-----|---------------------|
| Blip2 | 2.7B | ViT-B/L/g | OPT | 32 |
| InstructBlip | 7B | ViT | Vicuna | 32 |
| LLaVA-1.5 | 7B | CLIP ViT-L/14 | Vicuna | 576 |
| Qwen2-VL | 2B | EVA-CLIP ViT-L | Qwen-2 | dynamic |

the loss function. Then define $\mathcal{L}_{std}$ as:

$$\mathcal{L}_{std} = \sqrt{\frac{1}{m}\sum_{i=1}^{m}\left(\cos(\mathcal{M}(\mathcal{E}(x+\delta)[i], t_i) - \mathcal{L}_{sim}\right)^2}, \tag{7}$$

which is the standard deviation of the cosine similarity between the visual and text embeddings. Based on the two loss terms, the optimization objective is formulated as:

$$\min_{x^*} \mathcal{L}_{total} = -\mathcal{L}_{cos} + \alpha \cdot \mathcal{L}_{std}, \quad \text{s.t. } \|x^* - x\|_p \leq \epsilon, \tag{8}$$

where $\alpha$ is a hyperparameter that balances the two losses. Furthermore, we use a momentum $\mu$ to control the update of $x^*$. The specific process is shown in Algorithm 1.

## 5 EXPERIMENTS

### 5.1 EXPERIMENTAL SETUPS

**Models and datasets.** This study employs four open-source models: Blip2, InstructBlip, LLaVA, and Qwen2-VL. Table 1 presents detailed information about these models. Unlike the first three, Qwen2-VL's number of visual tokens varies with image resolution. Consequently, prior to feeding images into Qwen2-VL, we uniformly resize them to $336 \times 336$, resulting in 144 visual tokens. In the Visual Question Answering task, Blip2 and InstructBlip use the language prompt "Please describe this picture. Answer:", whereas LLaVA and Qwen2-VL utilize a conversational template with the text portion "Please describe this picture." We randomly select 100 images from the MS-COCO dataset as experimental samples.

**Baselines and setups.** As a baseline, we use the original images, images with added random noise, and verbose images. The perturbation magnitude is set to $\epsilon = 8$ under an $\ell_\infty$ constraint. For both the verbose images and our method, we employ the PGD algorithm with 5,000 iterations. For the verbose images, the step size and momentum are set to 0.0039 and 0.9, respectively, as reported in the original source. For our method, the weight is $\alpha = 0.8$, the step size $(lr)$ is 0.0022, and the momentum is $\mu = 0.9$. In the reinforcement-learning component, we use PPO; the attacker LLM is GPT-2 XL with a learning rate of $1.46 \times 10^{-5}$ and a clip range of 0.3. After the attacker LLM generates a token sequence, we extract a slice and repeat that slice until it matches the number of visual tokens. For example, if the slice contains two tokens and the VLM (e.g., InstructBLIP) has 32 visual tokens, the slice is repeated $32/2$ times to match the visual-token count. For all VLMs used in our experiments, the maximum number of generated tokens is set to 1024, and generation is performed using greedy decoding.

**Evaluation metrics.** We record the number of tokens generated per image and compute the average generation length (Average length) across the 100 images, as well as the proportion of samples producing more than 1000 tokens (Extra long rate).

### 5.2 MAIN RESULTS

Table 2 presents the experimental results on four models. It can be seen that the number of generated tokens produced by images with added random noise is similar to that of the original images,

Table 2: Comparison of the text-generation induction effects (e.g., number of generated tokens) of the original images, images with added random noise, verbose images, and VTIA on Blip2, InstructBLIP, LLaVA, and Qwen2-VL.

| VLM model | Method | Average length | Average length / max length | Extra long rate (%) |
|-----------|--------|----------------|------------------------------|----------------------|
| Qwen2-VL | Origin | 158.14 | 0.1544 | 1 |
| | Noise | 145.04 | 0.1416 | 0 |
| | Verbose Images | 809.01 | 0.7900 | 70 |
| | **VTIA (ours)** | **1024** | **1.0000** | **100** |
| LLaVA | Origin | 108.38 | 0.1058 | 0 |
| | Noise | 108.58 | 0.1060 | 0 |
| | Verbose Images | 518.61 | 0.5065 | 42 |
| | **VTIA (ours)** | **1024** | **1.0000** | **100** |
| InstructBlip | Origin | 11.63 | 0.0114 | 0 |
| | Noise | 11.37 | 0.0111 | 0 |
| | Verbose Images | 1003.86 | 0.9803 | 98 |
| | **VTIA (ours)** | **1014** | **0.9902** | **99** |
| Blip2 | Origin | 8.4 | 0.0082 | 0 |
| | Noise | 8.32 | 0.0081 | 0 |
| | Verbose Images | 933.19 | 0.9113 | 91 |
| | **VTIA (ours)** | **1024** | **1.0000** | **100** |

Table 3: Ablation experiments on the four VLMs, comparing attack performance when the $\mathcal{L}_{std}$ term is included or excluded and when momentum is used or not.

| VLM model | $\mathcal{L}_{std}$ | With Momentum | | Without Momentum | |
|-----------|---------------------|----------------|---------------------|-------------------|---------------------|
| | | Average length | Extra long rate (%) | Average length | Extra long rate (%) |
| Qwen2-VL | ✓ | 1024 | 100 | 1024 | 100 |
| | ✗ | 1024 | 100 | 1010.75 | 98 |
| LLaVA | ✓ | 1024 | 100 | 1021.31 | 99 |
| | ✗ | 1023.76 | 100 | 1023.81 | 100 |
| InstructBlip | ✓ | 1014 | 99 | 793.66 | 77 |
| | ✗ | 1004.27 | 98 | 551.06 | 53 |
| Blip2 | ✓ | 1024 | 100 | 902.9 | 88 |
| | ✗ | 994.18 | 97 | 640.34 | 62 |

indicating that simply adding random noise is insufficient to trigger verbose outputs from VLMs; achieving verbose outputs requires carefully designed image perturbations. Although the verbose images method can generate malicious images that induce verbose text, its effectiveness remains inferior to our proposed method. The performance gap is especially pronounced for the two more recent models, LLaVA and Qwen2-VL (Gao et al. (2024a) did not evaluate these two models), which further demonstrates the advantage of the "search adversarial prompt first, then optimize image perturbations" strategy.

Figure 3 displays the original images and the adversarial images, and compares the cosine-similarity distributions between their visual embeddings and the embeddings of the adversarial prompt. The results show that after applying small perturbations to the original images, the cosine similarities of most visual embeddings with their corresponding adversarial-prompt embeddings increase. Consequently, the perturbed images become semantically closer to the adversarial prompt and can trigger verbose outputs from the VLM in the same way as that prompt.

## 5.3 ABLATION STUDIES

We primarily investigate the effects of the $\mathcal{L}_{std}$ term, hyperparameter $\alpha$, momentum, perturbation magnitude, and the adversarial prompt on attack performance.

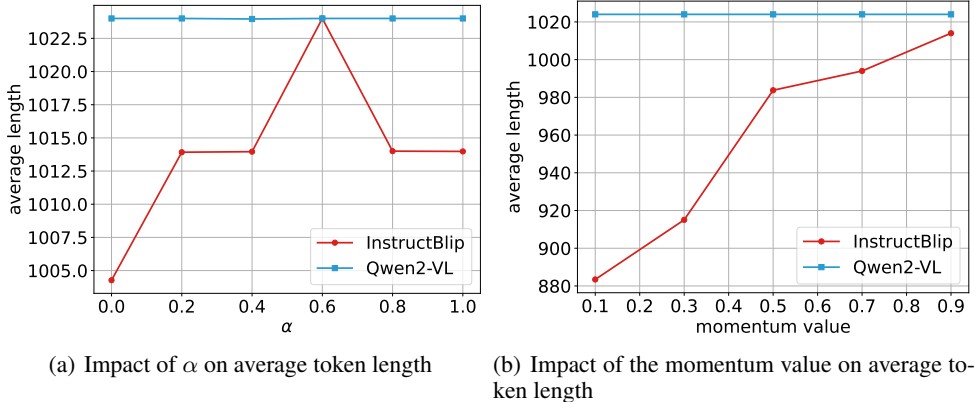

(a) Original images, adversarial images, and their generated outputs

(b) Cosine similarity of origin image

(c) Cosine similarity of adversarial image

Figure 3: Examples of original images and adversarial images, together with distributions of the cosine similarity between their visual embeddings and the adversarial-prompt embeddings. The model used is LLaVA.

(a) Impact of $\alpha$ on average token length

(b) Impact of the momentum value on average token length

Figure 4: Effect of different weights $\alpha$ and different momentum values on attack performance (e.g., average generated token length), shown as curves or bar charts.

**Impact of the $\mathcal{L}_{std}$ term.** Table 3 reports ablation experiments on the $\mathcal{L}_{std}$ term and momentum. Vertical comparisons indicate that adding $\mathcal{L}_{std}$ to the loss improves attack performance, particularly on Blip2 and InstructBlip; this improvement is more pronounced when momentum is not used. For LLaVA and Qwen2-VL, the impact of including $\mathcal{L}_{std}$ is relatively small; moreover, when momentum is absent, adding $\mathcal{L}_{std}$ slightly degrades LLaVA's attack performance. Two main reasons explain this phenomenon: 1) LLaVA and Qwen2-VL have far more visual tokens than Blip2 and InstructBlip; therefore, even if some visual embeddings and their corresponding adversarial prompt embeddings are not fully optimized in terms of cosine similarity, the overall attack is less affected. When the number of visual tokens is small, such insufficiently optimized embeddings can substantially reduce attack effectiveness. 2) Introducing the $\mathcal{L}_{std}$ term into the loss can trade off optimization for the $\mathcal{L}_{sim}$ term when optimizing the adversarial image. This trade-off may reduce final performance, especially when the visual token count is very large (e.g., LLaVA has 576 tokens). Horizontal comparisons show that introducing momentum improves attack performance across models.

**Impact of $\alpha$ and momentum value.** In addition, Figure 4 illustrates the effects of different $\alpha$ values and different momentum values on attack performance. As shown in Figure 4(a), performance on InstructBLIP is optimal when $\alpha = 0.6$, further indicating that an appropriate choice of $\alpha$ is needed to

Table 4: Attack performance and perceptual-quality metrics (e.g., LPIPS) under different perturbation magnitudes (e.g., 2/255, 4/255, 8/255, 16/255).

| Magnitude | LPIPS | Average length | Extra long rate (%) |
|---|---|---|---|
| 2/255 | 0.0110 | 380.34 | 36 |
| 4/255 | 0.0379 | 793 | 77 |
| 8/255 | 0.1137 | 1014 | 99 |
| 16/255 | 0.2190 | 1024 | 100 |

Table 5: Attack performance under different adversarial prompts (constructed from repeated slices); the repetition count is computed as: visual token number/slice token number

| Slice length | Average length | Extra long rate (%) |
|---|---|---|
| 2 | 1014 | 99 |
| 4 | 1003.91 | 98 |
| 8 | 923.44 | 90 |
| 16 | 953.55 | 93 |
| 32 | 883.48 | 86 |

balance the optimization of $\mathcal{L}_{sim}$ and $\mathcal{L}_{std}$. In contrast, on Qwen2-VL, $\alpha$ has little impact on attack performance, which corroborates that when the number of visual tokens is large, whether each visual embedding is fully optimized has a reduced influence on the final outcome. Figure 4(b) shows that, for InstructBLIP, attack performance increases as the momentum value grows, implying that stable optimization is necessary for generating effective adversarial images; whereas for Qwen2-VL, the momentum value has little effect, likely because a larger number of visual tokens makes the overall optimization process more stable.

**Impact of $\epsilon$.** Table 4 compares the impact of the perturbation magnitude $\epsilon$ (2/255, 4/255, 8/255, 16/255) on attack performance and reports the LPIPS between adversarial and source images. The results show that attack strength increases significantly with larger perturbation magnitude, but the perturbations also become more perceptible. Therefore, in practical attacks one must trade off stealthiness and attack effectiveness and select an appropriate perturbation magnitude.

**Impact of slice length.** Table 5 presents the impact of different slice lengths on attack performance. The table shows that although various adversarial prompts can all induce VLMs to generate tokens up to the maximum limit, the final attack effectiveness of the resulting adversarial images still differs after the vision-aligned perturbation optimization step. Moreover, as the slice length increases, attack performance tends to decline. We attribute this mainly to the large amount of repetition and redundancy in image pixels: if an adversarial prompt contains many repeated tokens, it matches the image's information-carrying characteristics and thereby reduces the difficulty of the vision-aligned perturbation optimization.

## 6 CONCLUSION

This paper aims to construct imperceptible image perturbations that induce VLMs to produce verbose responses, thereby increasing the computational, time, and monetary costs associated with the inference process of VLMs. To achieve this, we propose a two-stage decoupled attack, named VTIA. In stage one, we treat the VLM's generated token count as a reward and apply reinforcement learning to optimize an attacker LLM that discovers adversarial prompt embeddings. In stage two, we optimize image perturbations by the trade-off between the similarity loss and the standard deviation loss, ensuring that the visual embeddings align with the adversarial-prompt embeddings while keeping the perturbations visually imperceptible. Experiments on popular VLMs — BLIP2, InstructBLIP, LLaVA, and Qwen2-VL — show that the constructed adversarial images significantly increase the number of generated tokens while maintaining high visual stealthiness, highlighting the potential threat of such attacks in real-world deployments.

**Ethics statement.** This paper investigates the security vulnerabilities of VLMs by proposing a verbose-text induction attack that maliciously prolongs model outputs. Our goal is not to promote harmful usage but to highlight critical risks associated with excessive token generation, which can inflate energy consumption, increase operational costs, and impair system responsiveness. All experiments were conducted on publicly available models and datasets. No private or sensitive data was used, and no real-world deployment systems were attacked. We release our findings in the spirit of responsible disclosure, aiming to assist the community in understanding potential risks and motivating the development of more robust and cost-efficient VLMs.

**Reproducibility statement.** To ensure reproducibility, we provide comprehensive details of our methodology and experimental setup. Specifically, we describe the two-stage framework, including reinforcement learning strategies for adversarial prompt search and the vision-aligned perturbation optimization procedure. Hyperparameters, training configurations, and evaluation protocols are reported in the main paper and supplementary material. Experiments were conducted on four widely used VLMs with publicly available checkpoints. All code, configurations, and perturbation generation scripts will be released upon publication to facilitate verification and further research.

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

# Appendix

## A    USE OF LARGE LANGUAGE MODELS

In this study, large language models were used solely to polish the manuscript text, improving the fluency and clarity of the writing.

## B    EXPERIMENTAL DETAILS

The conversational template used by LLaVA and Qwen2-VL is as follows:

```
Template

conversation = [
    {

        "role": "user",
        "content": [
            {"type": "text", "text": "Please describe this picture."},
            {"type": "image"},
          ],
    },
]
```

## C    ADDITIONAL EXPERIMENTS

Figure 5 presents the token-length distributions of original images and adversarial images across four models. For all four models, the token lengths of original images are concentrated toward the left, whereas those of adversarial images cluster on the far right. Meanwhile, due to model-specific characteristics, LLAVA and Qwen2-VL generate more tokens than BLIP2 and InstructBLIP on original images, and their distributions are close to Gaussian.

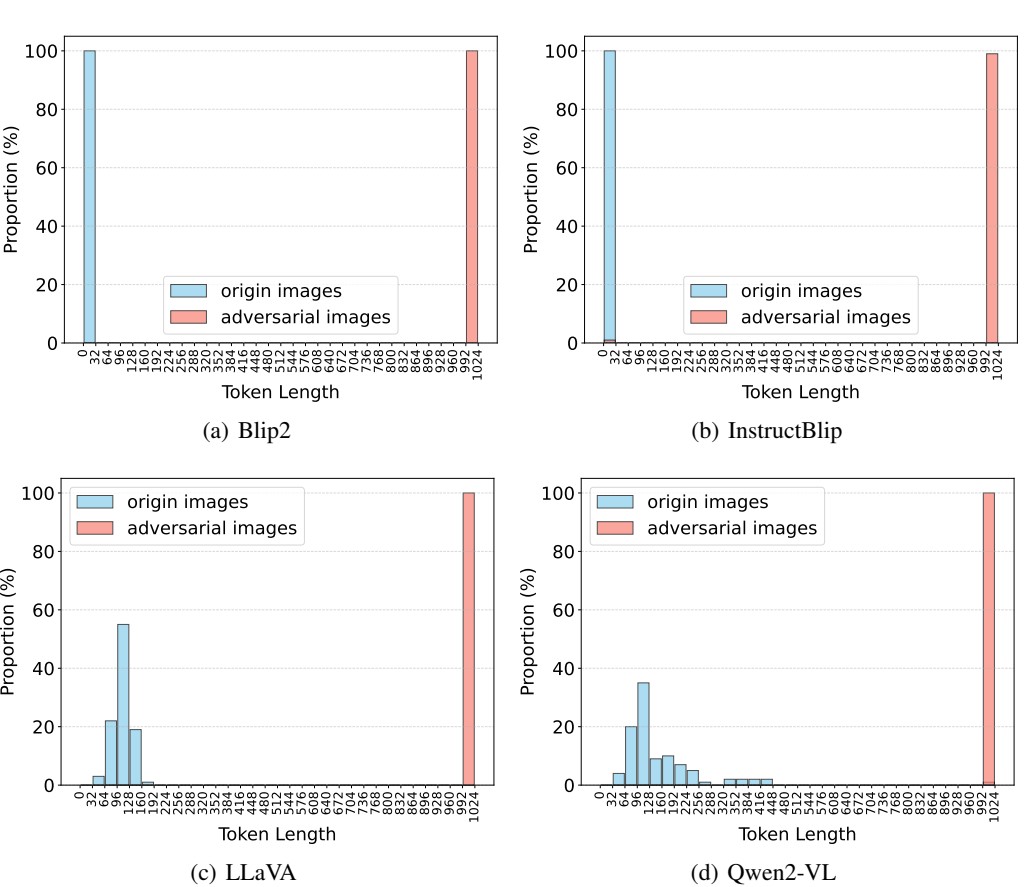

Figure 5: The token-length distribution of original images and adversarial images across the four models.

