# OpenReview forum: "An Image Is Worth Ten Thousand Words: Verbose-Text Induction Attacks on VLMs"
_ICLR.cc/2026/Conference — ICLR 2026 Conference Withdrawn Submission_

### Official Review · Reviewer_vrSH · 2025-10-25

**Soundness:** 3
**Presentation:** 2
**Contribution:** 3
**Rating:** 4
**Confidence:** 3

**Summary:**

This paper proposes a novel Verbose-Text Induction Attack (VTIA) targeting Vision-Language Models (VLMs). The attack aims to inject imperceptible perturbations into images so that the target VLMs generate excessively long (verbose) outputs, thereby increasing inference time, energy use, and token cost. Experiments on four representative VLMs (BLIP2, InstructBLIP, LLaVA, and Qwen2-VL) show that VTIA dramatically increases output lengths (up to 1000+ tokens) while maintaining high perceptual similarity (low LPIPS).

**Strengths:**

1. Novel attack objective.  The paper directly maximizes the number of output tokens as the attack goal, providing a more stable and controllable objective compared to prior methods.
2. Effective two-stage decoupled attack framework. The separation of prompt-level RL optimization and image-level perturbation optimization is technically sound.
3. State-of-the-art results.  The attack achieves state-of-the-art performance, significantly outperforming baseline methods across four VLMs.

**Weaknesses:**

1.  Evaluation is insufficient. The method is evaluated on only 100 images randomly selected from MS-COCO. This sample size is too small; at least 1,000 images should be used for a more reliable assessment. Additionally, evaluation across multiple datasets is necessary to demonstrate the method’s generalizability.
2. Ablation studies lack consistency. Some experiments are conducted on four VLMs, while others are limited to only two, making it difficult to fairly assess the contributions of each component.
3. Insufficient comparisons. The paper presents comparison results only with verbose images (Ref. 1) and does not include comparisons using verbose samples (Ref. 2).
Ref 1: Inducing high energy-latency of large vision-language models with verbose images. arXiv preprint arXiv:2401.11170, 2024a.
Ref 2: Energy-latency manipulation of multi-modal large language models via verbose samples. arXiv preprint arXiv:2404.16557, 2024b.
4. Limited discussion of real-world applicability. The attack is demonstrated in a controlled setting, but the paper lacks discussion of its feasibility under more restrictive black-box conditions or against deployed VLM services with additional safeguards, such as input filtering.
5. Minor issue: The equation in line 161 (the last line of page 3) appears to be incorrect and should be revised.

**Questions:**

1. Visual interpretation and attribution analysis. To better reveal the mechanisms behind the verbose-text induction attack, the authors should provide visual interpretations and token–region attribution analyses that highlight which image regions most strongly drive generation of long/verbose sequences. Such analyses will (1) offer mechanistic insight into why certain perturbations lengthen outputs, (2) help verify that the attack targets semantic content rather than spurious features, and (3) increase reproducibility and trust in the results.
2. Expanded performance metrics. Please report more evaluation metrics, such as inference latency and captioning quality, to provide a more comprehensive assessment of the method’s effectiveness.

---

### Official Review · Reviewer_1hH7 · 2025-10-27

**Soundness:** 1
**Presentation:** 2
**Contribution:** 1
**Rating:** 2
**Confidence:** 4

**Summary:**

This paper proposes a novel attack method named VTIA, Verbose-Text Induction Attack. The goal is to craft an adversarial image that appears benign but, when fed into a VLM, induces the model to generate an extremely long and low-information text response to increase the cost.

**Strengths:**

- The two-stage (search-then-align) methodological framework is clear and easy to understand.

- The method successfully induced the target open-source models to generate maximum-length tokens, demonstrating technical feasibility in the controlled setting.

**Weaknesses:**

- I cannot understand the fundamental attack scenario.
The paper claims the attack increases "user costs", which makes no sense if the user is the attacker, as they would just be increasing their own costs. This is completely different from general adversarial attacks (general adversarial attacks assue the user is the attacker).
 If the paper suggests a DoS attack against the service provider, this is an inefficient vector: an attacker could simply spam the API with normal requests to achieve the same goal much more easily.

- The work completely ignores simpler and more direct text-based attack vectors, such as using prompt engineering or **jailbreaking** to ask the LLM component, as the authors claim that "recent VLM relys on LLM". So the paper fails to compare its complex image-based attack against this obvious text-based baseline, undermining the necessity of the proposed method.

- The method has extremely low practical relevance because it relies on a "gray-box" threat model.
The attack requires access to the internal parameters of the visual encoder and intermediate modules, which is impossible for the commercial, black-box models (like GPT-4V, Gemini) that actually operate on a "per-token" cost basis.
The paper does not evaluate the transferability of the attack, further limiting its application to real-world scenarios.

**Questions:**

- Can you clarify a realistic and compelling attack scenario where an attacker must use a complex adversarial image, rather than simpler alternatives like spamming normal requests or using malicious text prompts?

- Why did the paper not include a comparison against the most obvious baseline: using text-based prompt engineering to induce verbose outputs from the LLM component?

- Given that this attack requires gray-box access, how can it be applied to the real-world, black-box commercial VLMs that are the primary examples of token-based billing? What is the practical security implication if it cannot be applied?

---

### Official Review · Reviewer_KcUd · 2025-11-01

**Soundness:** 3
**Presentation:** 3
**Contribution:** 2
**Rating:** 4
**Confidence:** 5

**Summary:**

This paper introduces a Verbose-Text Induction Attack (VTIA) that injects adversarial perturbations into images to induce vision-language models (VLMs) to generate excessively long textual outputs. The method explicitly optimizes for output token length rather than indirectly delaying the EOS token. VTIA uses a two-stage process: (1) adversarial prompt search via reinforcement learning to identify prompts that trigger verbosity, and (2) vision-aligned perturbation optimization to align image embeddings with adversarial prompt embeddings. Experiments on several popular VLMs show that VTIA can effectively increase output verbosity and demonstrate strong transferability.

**Strengths:**

1. For originality, this paper proposes a novel attack objective, which focuses on generation length and verbosity as a vulnerability metric. This is original and highlights an overlooked efficiency issue in VLM deployment.

2. For clarity, the proposed two-stage pipeline combining prompt search and image-space optimization is well-formulated and intuitive.

3. For quality, the evaluations across multiple VLMs provide a convincing demonstration of the generality of the method.

**Weaknesses:**

1. There is a lack of comparison with representative baselines. The paper does not compare against existing multimodal attack frameworks such as VLAttack, which limits understanding of its relative effectiveness.

2. The focused problem setting is incremental. Adversarial attacks on VLMs have been extensively studied, and the contribution mainly reorients the objective toward verbosity, which may be viewed as a narrow extension rather than a fundamentally new attack paradigm.

3. The quality of the generated text is missing. While the method maximizes output length, there is no quantitative analysis of the informational or semantic quality of the generated verbose text, which is important for assessing real-world impact.

4. There is only limited discussion on practicality.1 It remains unclear how such verbose induction affects model usability or downstream performance in realistic scenarios.

**Questions:**

1. Could the authors include a comparison with VLAttack or other vision-language adversarial methods to contextualize their improvement?

2. How is “verbose” output quality measured—could metrics like perplexity, redundancy, or semantic coherence be reported?

3. What is the computational overhead of the reinforcement learning–based prompt search stage?

4. Would combining verbosity objectives with traditional adversarial goals (e.g., misclassification or hallucination) yield stronger or more general attacks?

---

### Official Review · Reviewer_mpXp · 2025-11-07

**Soundness:** 2
**Presentation:** 2
**Contribution:** 2
**Rating:** 4
**Confidence:** 3

**Summary:**

The paper introduces the Verbose-Text Induction Attack (VTIA), a novel framework that combines reinforcement learning with vision-aligned perturbation optimization. It first generates an adversarial prompt embedding using reinforcement learning and then optimizes visual perturbations to align the image embeddings with this adversarial prompt, effectively inducing the Vision-Language Model (VLM) to generate excessively long responses. The authors evaluate VTIA on several models, including Blip2, InstructBlip, LLaVA, and Qwen2-VL. The results demonstrate VTIA's significant advantage in terms of the number of tokens generated and the rate of extra-long responses. Ablation studies further analyze the impact of individual components and hyperparameters on the attack's effectiveness.

**Strengths:**

-  The paper is well-organized and clearly structured.
- The paper proposes the VTIA attack framework, a novel methodology that integrates reinforcement learning with vision-aligned perturbation optimization. Its two-stage decoupled design ensures that the second phase operates completely independently of the target VLM’s textual module, thereby circumventing the significant computational overhead associated with repeatedly invoking large LLMs during iterative optimization.
- Experiments are conducted on multiple models such as Blip2, InstructBlip, LLaVA, and Qwen2-VL, demonstrating the superiority of the VTIA framework. Additionally, extensive ablation studies are designed to validate and analyze the impact of each component and associated hyperparameters on the method's performance.

**Weaknesses:**

- **Limited Evaluation Scope**: Although experiments are conducted on four models, the evaluation could be further strengthened by including a broader and more diverse set of modern VLMs (e.g., LLaVA-NEXT, Qwen3-VL) to better demonstrate generalizability. Moreover, the use of only 100 randomly selected images from the MSCOCO dataset constitutes a relatively small sample size, raising concerns about the robustness and real-world applicability of the proposed method. It is also suggested to evaluate the attack on additional tasks or benchmarks (e.g., visual question answering) to verify its generalizability across modalities and task settings.

- **Ambiguous Notation**: In Equation (8), the term $L_{cos}$ is used without definition. It remains unclear whether this corresponds to $L_{sim}$ from Equation (6).

- **Lack of Robustness Analysis**: The paper does not assess the robustness of the proposed method against common defense strategies. It is recommended to analyze its effectiveness under mainstream defenses such as smoothing-based methods (e.g., Gaussian, average, and median smoothing), GAN-based defenses (e.g., NRP [1]), and diffusion-based approaches (e.g., DiffPure [2]).


- **Max Token Limitation**: The maximum token length is set to 1024, and most samples reach this limit. The authors should consider increasing the maximum token length to better showcase the true effectiveness of the adversarial attack and avoid potential truncation effects.

[1] M Naseer et al. A self-supervised approach for adversarial robustness. In CVPR 2020

[2]W Nie et al. Diffusion models for adversarial purification. In ICML 2022

**Questions:**

- **Transferability**: How transferable is the attack? Specifically, does the adversarial perturbation remain effective when applied to different images or prompts?
- **Semantic Interpretation**: Does the adversarial prompt convey meaningful semantic information? It would be helpful to include a visualization of the generated prompt to illustrate its interpretability and semantic coherence.
- **Text-Image Consistency**:
What is the embedding similarity between the attacked image and the generated text? Does the generated text deviate significantly from the image semantics? Additionally, it is suggested to provide quantitative metrics of text quality (e.g., BLEU, perplexity, or GPT score) to evaluate robustness against similarity-based or text-quality-based detection methods.

---

### Note · Authors · 2025-12-08

I have read and agree with the venue's withdrawal policy on behalf of myself and my co-authors.